# Altered Genome-Wide DNA Methylation in Peripheral Blood of South African Women with Gestational Diabetes Mellitus

**DOI:** 10.3390/ijms20235828

**Published:** 2019-11-20

**Authors:** Stephanie Dias, Sumaiya Adam, Paul Rheeder, Johan Louw, Carmen Pheiffer

**Affiliations:** 1Biomedical Research and Innovation Platform (BRIP), South African Medical Research Council, P.O. Box 19070, Tygerberg, Cape Town 7505, South Africa; Stephanie.Dias@mrc.ac.za (S.D.); Johan.Louw@mrc.ac.za (J.L.); 2Department of Obstetrics and Gynecology, University of Pretoria, Private Bag X169, Pretoria 0001, South Africa; sumaiya.adam@up.ac.za; 3Department of Internal Medicine, Faculty of Health Sciences, University of Pretoria, Private Bag X169, Pretoria 0001, South Africa; paul.rheeder@up.ac.za; 4Department of Biochemistry and Microbiology, University of Zululand, Private Bag X1001, Kwa-Dlangezwa 3886, South Africa; 5Division of Medical Physiology, Faculty of Health Sciences, Stellenbosch University, P.O. Box 19063, Tygerberg, Cape Town 7505, South Africa

**Keywords:** gestational diabetes mellitus, molecular biomarkers, DNA methylation, MethylationEPIC Bead Chip Array, South Africa

## Abstract

Increasing evidence implicate altered DNA methylation in the pathophysiology of gestational diabetes mellitus (GDM). This exploratory study probed the association between GDM and peripheral blood DNA methylation patterns in South African women. Genome-wide DNA methylation profiling was conducted in women with (*n* = 12) or without (*n* = 12) GDM using the Illumina Infinium HumanMethylationEPIC BeadChip array. Functional analysis of differentially methylated genes was conducted using Gene Ontology (GO) and Kyoto Encyclopedia of Genes and Genomes (KEGG) pathway analyses. A total of 1046 CpG sites (associated with 939 genes) were differentially methylated between GDM and non-GDM groups. Enriched pathways included GDM-related pathways such as insulin resistance, glucose metabolism and inflammation. DNA methylation of the top five CpG loci showed distinct methylation patterns in GDM and non-GDM groups and was correlated with glucose concentrations. Of these, one CpG site mapped to the calmodulin-binding transcription activator 1 (*CAMTA1*) gene, which have been shown to regulate insulin production and secretion and may offer potential as an epigenetic biomarker in our population. Further validation using pyrosequencing and conducting longitudinal studies in large sample sizes and in different populations are required to investigate their candidacy as biomarkers of GDM.

## 1. Introduction

Gestational diabetes mellitus (GDM) is defined as glucose intolerance that arises during pregnancy, and usually resolves postpartum. The prevalence of GDM is increasing, affecting approximately 14% of pregnancies globally [1], although rates vary between <1% and 28% according to the diagnostic criteria employed and population studied [2]. GDM is associated with maternal (preeclampsia, caesarean section and birth injuries), fetal (macrosomia, shoulder dystocia, hyperinsulinemia, hypoglycemia, hyperbilirubinemia) and perinatal (respiratory distress syndrome, metabolic derangements and jaundice) complications [3,4,5], while both mothers and their offspring are at an increased risk of developing metabolic disease in later life [6,7,8]. Current estimates indicate that more than 50% of women with GDM develop type 2 diabetes (T2D) within 10 years, making GDM a strong predictor of T2D [6,9]. The identification of women with GDM who are at risk of developing T2D allows the introduction of timely measures to prevent or better manage disease progression. 

Epigenetic mechanisms are increasingly being implicated in the pathophysiology of metabolic diseases, including GDM [10]. DNA methylation, the most widely studied and best characterized epigenetic marker, is a reversible process that refers to the addition of a methyl group to the fifth carbon position of a cytosine residue within a cytosine-phosphate-guanine (CpG) dinucleotide, and regulates gene expression through transcriptional mechanisms [11]. Altered global and gene-specific DNA methylation are observed in the placenta of women with GDM [12,13]. DNA methylation is a tissue-specific process, although recent evidence suggests that peripheral blood reflects DNA methylation in tissue [14], while several studies report that maternal blood reflects pregnancy-associated DNA methylation changes [15,16,17], supporting its potential as epigenetic biomarkers for GDM. 

DNA methylation during GDM has been studied using various techniques such as enzyme-linked immunosorbent assays, whole-genome bisulfite sequencing, methylated DNA immunoprecipitation sequencing, liquid chromatography coupled with mass spectrometry, pyrosequencing, bead chip arrays and methyl light polymerase chain reaction (PCR) [12,15,17,18,19,20]. Due to its comparatively low cost compared to sequencing, reproducibility and high sample throughput, bead chip arrays are currently the most widely used technique for genome-wide DNA methylation profiling [21,22]. The current bead chip array version, the HumanMethylationEPIC, allows the interrogation of >850,000 CpG sites across the genome, enriched for promoters and enhancer sequences, covering 99% of RefSeq genes [23]. Previous versions, the HumanMethylation450 and HumanMethylation27, measured >480,000 and >27,000 CpG sites, respectively across the genome [21]. 

In South Africa, the prevalence of GDM has increased from about 1.6–25.8% in recent years [24,25]. The possible increase in future T2D cases will place a major burden on the already overburdened health system and creates an urgent need to identify preventative strategies. DNA methylation has attracted considerable interest as biomarkers that could facilitate risk stratification and offer opportunities for intervention strategies to prevent or delay the development of T2D after pregnancy [26]. The aim of this study is to explore the potential of DNA methylation to serve as biomarkers of GDM in black South African women. Genome-wide DNA methylation profiling was conducted in the peripheral blood of women with (*n* = 12) or without (*n* = 12) GDM using the Illumina methylationEPIC Bead Chip array. Functional analysis of differentially methylated genes was conducted to identify pathways associated with GDM in the South African population.

## 2. Results

### 2.1. Study Participants

Participant characteristics are presented in Table 1. As expected, no difference in age, gestational age and body mass index (BMI) was observed between women with or without GDM. Women with GDM had significantly higher fasting (*p* < 0.001) and 1 h oral glucose tolerance test (OGTT) (*p* < 0.01) glucose concentrations compared to women without GDM, while 2 h OGTT (*p* = 0.07) glucose concentrations showed a trend towards significance. In addition, fasting insulin concentrations, homeostatic model of assessment (HOMA), and c-reactive protein (CRP) levels were higher in women with GDM compared to women without GDM, although these were not statistically significant. No difference between groups were observed for HbA1c and adiponectin concentrations, nor for common risk factors (advanced maternal age (age ≥ 35 years), obesity (BMI ≥ 30 kg/m^2^), family history of diabetes mellitus, delivery of a previous baby more than four kg, glucosuria, previous recurrent pregnancy loss, stillbirth, or birth of a baby with congenital abnormalities), as well as education and employment status. 

### 2.2. Genome-Wide DNA Methylation Profiling

The average detection *p*-values for all probes were calculated for each sample and are presented in Appendix A. Each sample showed *p*-values below the usual cut-off of 0.01, indicating that all samples passed the quality control. In addition, box and whisker plots showed concordance across samples without any outliers, suggesting good quality and consistency of samples (Figure 1). Median β-values ranged between 0.79 and 0.83 across the 24 samples. A histogram of β-values showing the frequency distribution of CpG methylation across all samples is illustrated in Appendix A. A clear separation between GDM and non-GDM groups is evident in the principal component analysis (PCA) score plot, with characteristic DNA methylation profiles aggregating together within the same group (Figure 2). The first three PCAs explain 27.6% of the variance observed. The β-values were then converted to M-values for statistical analysis. To identify differentially methylated CpG sites between GDM and non-GDM pregnancies, data were filtered using the criteria shown in Figure 3. An M-value cut-off threshold between >0.4 and >0.6 was explored in this study, which is within the threshold range suggested by Du et al. [27]. In the first filtering step a M-value difference of >0.4 or <−0.4 and unadjusted *p* < 0.01 was used, to permit comparison between differentially methylated probes. Further filtering steps including M-values which ranged between >0.5 or <−0.5 and >0.6 or <−0.6 with unadjusted *p* < 0.01 were assessed. We identified 1046 differentially methylated CpG loci with M-value differences of >0.6 or <−0.6 and unadjusted *p* < 0.01 (Appendix A). To facilitate a more stringent analysis, a false discovery rate (FDR) <0.1 was added, which did not identify any significant probes. Hierarchical clustering was performed to determine whether these methylation patterns could distinguish between women with or without GDM. The heatmap in Figure 4 illustrates that there are distinct methylation patterns between the GDM and non-GDM groups. 

Of the 1046 differentially methylated CpG loci, 148 CpG sites (14.2%) were hypermethylated and 898 CpG sites (85.8%) were hypomethylated in women with GDM compared to women without GDM. To increase the likelihood of identifying differentially methylated promoters, probes located 5 kbp upstream or up to 3 kbp downstream of the transcription start site were also included as promoter regions. The frequency of all CpG sites analysed and differentially methylated CpG sites in relation to their genomic location is shown in Figure 5. Of the differentially methylated CpGs, 16.3% were associated with 5’-untranslated regions (UTR), 49.7% with promoters, 6.2% with coding domain sequences (CDS), 19.1% with introns, 4.0% with non-coding regions, 2.1% with 3’-UTRs and 4.6% with intergenic regions. Differentially methylated CpG sites were annotated to 939 unique genes using RefSeq build 87 (Appendix A). The top five significantly differentially methylated CpG sites selected for further analysis, were associated with four unique genes, including Solute Carrier Family 9 Member A3 *(SLC9A3)*, Male-Enhanced Antigen 1; Kelch domain-containing protein 3 *(MEA1;KLHDC3)*, Calmodulin Binding Transcription Activator 1 *(CAMTA1)* and RAS P21 Protein Activator 3 *(RASA3)*, and one unknown gene. The probe ID, location, gene region and direction of methylation (GDM vs. non-GDM), as well as the nearest gene/regulatory region for the unknown gene is shown in Table 2. Of the differentially methylated CpG sites, cg22985016 and cg16306629 was shown to be significantly hypermethylated, while cg21910650, cg23643951 and cg07966372 was significantly hypomethylated in GDM compared to non-GDM groups. The association between GDM and the top five CpG sites remained significant for each CpG after linear regression adjusting for age BMI and gestational age (Table 3). To examine the degree to which DNA methylation levels at these CpGs are associated with the clinical characteristics of GDM, Pearson’s correlation analysis was performed (Table 4). For cg22985016 and cg16306629, a positive correlation between DNA methylation and fasting glucose concentrations was observed, while methylation at cg21910650, g23643951 and cg07966372 was inversely correlated with glucose concentrations. Furthermore, DNA methylation at cg22985016 and cg16306629 was correlated with 1 h glucose, while methylation at cg07966372 was negatively correlated with fasting insulin concentrations. When adjusting for GDM, the association between the five CpGs and fasting glucose concentrations and between cg22985016 and cg16306629 and 1 h OGTT was no longer significant, while the association between cg07966372 and fasting insulin remained significant (Appendix A). 

### 2.3. Functional Enrichment Analysis

Differentially methylated CpG sites (1046), annotated to 939 unique genes using M-values >0.6 and <–06 with unadjusted *p* < 0.01 threshold criteria, were selected for functional enrichment analysis. Functional enrichment analysis identified 261 *Kyoto Encyclopedia of Genes and Genomes* (KEGG) pathways, including pathways for T2D and insulin signaling (Appendix A). Only 50 KEGG pathways were statistically significantly different between GDM and non-GDM groups (Appendix A). Statistically significant pathways included cancer, brain signaling, cell growth, proliferation, viability and inflammation pathways. The most significant KEGG pathway was ‘Signaling pathways regulating pluripotency of stem cells’ with an enrichment score of 10.496, a *p*-value = 2.76 × 10^−5^ and 19 differentially methylated associated genes. In addition, Gene Ontology (GO) terms were enriched by differentially methylated genes, categorized into 1181 biological processes, 167 molecular functions and 85 cellular components with a *p*-value < 0.05 (Appendix A). The top 10 GO terms categorized into biological processes, molecular functions and cellular components are illustrated in Figure 6. Of these, homophilic cell adhesion via plasma membrane adhesion molecules (biological process), calcium ion binding (molecular function) and integral component of plasma membrane (cellular component) have the highest ranked enrichment score and *p*-value < 0.001. 

## 3. Discussion

We report the differential methylation of 1046 CpG sites in the peripheral blood of black South African women with GDM compared to women with normoglycemic pregnancies. Functional analysis mapped these CpGs to genes in pathways key to metabolic regulation. Furthermore, differential methylation of the five CpG loci, within *SLC93A* was positively correlated with fasting and 1 h glucose, while CpGs within *CAMTA, MEA1;KLHDC3* and *RASA3* was inversely correlated to fasting glucose, with distinct methylation profiles in GDM and non-GDM groups. *CAMTA1* is a transcriptional activator that was previously shown to regulate insulin production and secretion [28]. These results support the plausibility of the observed DNA methylation differences in GDM pathophysiology and potential as diagnostic biomarkers of GDM. 

Genome-wide DNA methylation differences during GDM have been demonstrated in other populations. Kang et al. used the Illumina Infinium Human MethylationEPIC Bead Chip array to investigate DNA methylation in Chinese women with GDM, and showed that the top 200 differentially methylated loci mapped to 151 genes [15]. Of these, 15 genes, *CAMTA1,* Smad Nuclear Interacting Protein 1 *(SNIP1)*, Protein-Tyrosine Phosphatase, Receptor-Type, F Polypeptide-Interacting Protein-Binding Protein 2 *(PPFIBP2),* Switching B Cell Complex Subunit SWAP70 (*SWAP70)*, Semiphorin 6D *(SEMA6D)*, Cadherin 8 *(CDH8)*, Cytochrome P450 Family 26 Subfamily B Member 1 *(CYP26B1),* Wnt Family Member 6 *(WNT6)*, Raftlin, Lipid Raft Linker 1 *(RFTN1),* Unc-5 Netrin Receptor C *(UNC5C),* Nucleoside Diphosphate-Linked Moiety X Motif 6 *(NUDT6),* Storkhead Box *(STOX2)*, MutS Protein Homolog 5 *(MSH5),* KH RNA Binding Domain Containing, Signal Transduction Associated 2 *(KHDRBS2),* and Neuregulin 1 *(NRG1)* were similarly shown to be differentially methylated in our study, and has been illustrated in a venn diagram (Appendix A). Disparities in the number of differentially methylated CpG sites identified between studies could be due to population differences such as ethnicity, age and stage of pregnancy, and the data filtering criteria used. Although M-values were used to measure methylation differences in both studies, Kang et al. used a more stringent FDR adjusted *p*-value < 0.05 for their analysis whereas we used an unadjusted *p*-value < 0.01, since an FDR of <0.05 did not identify any significantly differentially methylated loci in our analysis. Despite using a higher FDR than Kang et al., the differential methylation of 15 genes were similar between studies [15]. Other technical differences between studies which may affect methylation levels include sample preparation, loading during hybridization and batch effect bias [21,29]. Soriano-Tárraga et al. reported that the method of DNA extraction affects global DNA methylation levels [29]. Thus, standardization of analytical methods across laboratories is essential to enable comparison of DNA methylation patterns between studies. Other studies that used previous versions of the bead chip array similarly reported DNA methylation differences during GDM in Non-Hispanic Caucasian American and Caucasian English populations [16,17]. As reported in these studies [15,16,17,30], the majority of CpG differences in our study were hypomethylated in women with GDM compared to women without GDM. However, in contradiction, in our study most of the 1046 differentially methylated CpG sites occurred in promoter regions, whereas previous studies identified most of the differentially methylated CpGs in gene body regions [30,31]. Differences could be due to the method of analysis used. Our analysis included additional CpGs located 5 kbp upstream and 3 kbp downstream of the transcription start site to increase the probability of detecting differentially methylated promoter regions. Altered DNA methylation in promoter regions influences the expression of specific genes [13,32,33], which may enable the identification of genes/pathways involved in metabolic processes during GDM. 

Recently, we demonstrated that global DNA methylation is not associated with GDM in South African women [19]. We hypothesized that the failure to detect DNA methylation differences was due to technical limitations and that gene-specific methylation analysis would be able to identify GDM-associated methylation differences. Global DNA methylation quantification is a crude marker of overall genomic methylation and does not have the resolution to detect gene-specific differences, as observed in the current study. Similar findings were reported by Matsha et al., who showed no difference in global DNA methylation between 61 diabetic individuals on treatment and 287 normoglycemic subjects in a mixed ethnic ancestry South African population [34]. In addition, no difference in global DNA methylation was observed in peripheral blood mononuclear cells of a Danish population with obesity or T2D compared to controls [35]. 

The diagnosis of GDM is contentious and varies across countries and health institutions. Currently the International Association of Diabetes in Pregnancy Study Group (IADPSG) criteria are advocated by several international bodies and endorsed by the World Health Organisation (WHO) [36]. However, concerns that the high costs and increased workload of IADPSG criteria outweigh the clinical effects of small glucose differences has hampered its universal use. We were able to see altered DNA methylation patterns despite small glucose differences between women with or without GDM, suggesting that epigenetic programming is evident even during mild hyperglycemia. Kang et al. also demonstrated altered DNA methylation in women diagnosed with GDM according to IADPSG diagnostic criteria [15]. These findings support The Hyperglycemia and Adverse Pregnancy Outcomes (HAPO) study, which showed that even mild hyperglycemia is associated with adverse pregnancy outcomes and requires treatment [37]. Furthermore, several clinical trials have confirmed that treatment of mild hyperglycemia decreases maternal morbidity and adverse perinatal outcomes [38]. 

Functional analysis of differentially methylated CpG sites identified canonical pathways related to signal transduction, cell growth, proliferation, differentiation and apoptosis, insulin resistance, glucose metabolism, inflammation, neurological signaling, and oncogenesis. Altered DNA methylation of two signaling pathways, mitogen-activated protein kinase *(MAPK)* and phosphoinositide 3-kinase *(PI3K)*, which play a role in cell growth and differentiation, and the metabolic action of insulin [39], have previously been reported during GDM in other populations [15], identifying these CpG sites as likely biomarkers for the development of GDM. Our results demonstrated that pathways associated with cancer are differentially methylated in women with GDM compared to controls. Several studies have reported a link between GDM and cancer, particularly breast cancer [40,41,42], identifying GDM as a potential risk factor for the development of cancer in later life, Nine of the top 10 GO terms enriched for biological processes were associated with structural organization and developmental processes, supporting the influence of GDM on in utero programming of fetal growth and development [43]. As expected, all 10 GO terms enriched for molecular functions were associated with regulatory or binding activities and offer insight into functions influenced by altered methylation at a molecular level during GDM. 

A strength of our study is that women were matched for age, gestational age and BMI, to ensure that results were comparable between groups. In addition, DNA methylation analysis was conducted using the most comprehensive MethylationEPIC Bead Chip array currently available, which is considered a high-throughput method, that has a lower cost compared to sequencing, and is reproducible and time-efficient [21,22]. Our study has a number of limitations. The sample size (*n* = 24) is small, although, it is larger than previously reported [15,16,17]. No CpG sites reached FDR cut-off, suggesting that the study might have been underpowered. However, 15 of the differentially methylated genes identified in our study were amongst the top 151 identified by Kang et al. Peripheral blood cells consist of a mixture of different cell types [44], which may confound methylation analysis. In our study, cell type composition did not differ significantly between GDM and non-GDM groups and therefore was not adjusted for in further analysis due to the small sample size. Thus, methylation differences between cell types could have confounded our analysis. Furthermore, physical activity, diet, smoking and alcohol consumption, which are known to influence DNA methylation patterns, are not known, and could confound our analysis. However, women in our study were recruited from the same community and had similar lifestyle behaviours, education and employment status, suggesting that they had roughly similar environmental influences.

To our knowledge, this exploratory study is the first to profile genome-wide DNA methylation levels in the peripheral blood of South African women with GDM. We have identified five CpGs which are associated with GDM and offer potential as epigenetic biomarkers in our population. Further validation using pyrosequencing and conducting longitudinal studies in large sample sizes and in different populations are required to investigate their candidacy as biomarkers of GDM.

## 4. Materials and Methods

### 4.1. Study Participants

Ethical approval for this study was granted by the University of Pretoria Health Sciences Ethics Committee (180/2012: approved on the 26/09/2012). The study was conducted according to the Declaration of Helsinki and all women gave written informed voluntary consent after the procedures had been fully explained in the language of their choice. One thousand pregnant women attending a primary care clinic in Johannesburg, South Africa were enrolled in the study. At recruitment, demographic and socio-economic data were obtained in the form of a standardized questionnaire and risk factors for GDM, i.e. advanced maternal age (age ≥ 35 years), obesity (BMI ≥ 30 kg/m^2^), family history of diabetes mellitus, delivery of a previous baby more than four kilograms, glucosuria, previous recurrent pregnancy loss, stillbirth, or birth of a baby with congenital abnormalities) were assessed [25]. Patients with pre-existing diabetes mellitus (Type 1 diabetes (T1D) and T2D) and those who were more than 26 weeks pregnant were excluded. At their first visit, random glucose and glycated hemoglobin (HbA1c) concentrations were measured. Women with random glucose and HbA1c concentrations less than 11.1 mmol/L and 6.5 %, respectively, were requested to fast overnight and return to the clinic within two weeks. At this time, a 75 g oral glucose tolerance test (OGTT) was conducted, and GDM was diagnosed if at least one glucose value was met (fasting plasma glucose > 5.1 mmol/L, 1 h OGTT > 10 mmol/L or 2 h OGTT > 8.5 mmol/L), according to the IADPSG criteria [45]. Blood for measurement of cytokines and DNA methylation was collected and stored at –80 °C. For this sub-study, a subset of women with (*n* = 12) and without (*n* = 12) GDM were selected for genome-wide DNA methylation analysis. The inclusion criteria were pregnant women ≥18 and ≤40 years of age, black ethnicity, human immunodeficiency virus (HIV) negative and women with a singleton pregnancy. All women were matched according to age, BMI and gestational age as far as possible.

### 4.2. DNA Extraction

Genomic DNA was extracted from 2 ml of peripheral blood collected in Ethylenediaminetetraacetic acid (EDTA) tubes using the QIAamp DNA Blood Midi Kit (Qiagen, Hilden, North Rine-Westphalia, Germany), as previously described [19]. Briefly, white blood cells were lysed and loaded onto the QIAamp Midi column, bound DNA was washed and then eluted from the column membrane using 300 µl of elution buffer and centrifuged at 4500× *g* for 2 mins. DNA concentration was measured using the Qubit Fluorometer (Invitrogen, Carlsbad, CA, USA) and the Quanti-iT dsDNA Broad Range assay kit (ThermoFisher, Waltham, MA, USA). One microgram of DNA in a volume of 45 µl was frozen and shipped on dry ice, as instructed by the University of Southern California Molecular Genomics Core for genome-wide DNA methylation analysis using the Illumina Infinium HumanMethylationEPIC BeadChip (USC Molecular Genomics Core, Los Angeles, CA, USA). 

### 4.3. Genome-Wide DNA Methylation Profiling

Genome-wide DNA methylation profiling was conducted using the Illumina’s Infinium HumanMethylationEPIC Bead Chip (HumanMethylationEPIC, Illumina inc., San Diego, CA, USA) according to manufacturer’s instructions. Bisulfite conversion of 500 ng genomic DNA was performed using the Illumina-specific EZ DNA methylation kit (D5001, Zymo Research, Orange, CA, USA), and quality control was conducted by quantitative real-time polymerase chain reaction (PCR) and melt curve analysis. Bisulfite converted DNA was amplified up to 1000-fold with DNA polymerase during the incubation step in the Illumina hybridization oven at 37 °C. Amplicons were then fragmented to 300–600 bp products, precipitated with isopropanol and loaded onto Illumina Infinium HumanMethylationEPIC Bead Chips prepared for hybridization in the capillary flow-through chamber (Human MethylationEPIC, Illumina Inc.), according to the Infinium protocol [46]. After annealing to locus-specific 50-mer probes, a single base extension occurs at the base immediately adjacent to the interrogated CpG site. Products were fluorescently labelled with either dinitrophenol-labelled ddATP/ddTTP or biotin-labelled ddCTP/ddGTP, depending on the methylation state of the interrogated CpG site. Fluorescence intensity was measured with the Illumina iScan system (iScan Control Software v.3.3.28) and was based on the ratio of methylated probe intensities and the overall intensity (sum of methylated and unmethylated probe intensities). The methylation scores were represented as raw beta (β)-values and were exported as 48 intensity data files (IDAT). 

### 4.4. Processing and Analysis of the Human Methylation EPIC Bead Chip Array

Data analysis was conducted by Partek (Partek, St. Louis, MO, USA). IDAT files were imported to Partek (R) Genomics Suite (R) v.7.18.0803 software. Functional normalization with normal-exponential out-of-band (NOOB) background correction and dye correction was used [47]. Quality control was performed across all imported probes (865,859) for each sample. All samples passed the quality control, and those with detection *p* < 0.01 were included in the analysis. Thereafter, β-values for imported probes were plotted and no outliers were detected, indicating that the data were technically sound. In addition, a histogram was used to illustrate distribution of methylation β-values across all CpG sites in each sample. Data filtering was conducted to remove polymorphic probes (*n* = 22,139), cross-hybridising probes (*n* = 40,762), non-CpG probes (*n* = 1) and probes overlapping both the polymorphic and cross-hybridising probe lists (*n* = 1,721) (Figure 3), according to McCartney et al. [23]. The clean data set consisted of 801,236 probes (referred to as CpG sites). Exploratory analysis was performed using PCA. Cell count estimation was performed empirically using methylation data from sorted blood cells using the ‘Estimate Cell Count’ function in the minfi package in R [48]. The function is based on a modification of the original method by Houseman et al. [49] and the R package FlowSorted.Blood.450k [50]. No differences in cell composition were identified, and cell composition was deemed unlikely to be a confounder (Appendix A). Therefore, cell composition was not corrected for in further analysis. 

Following data processing, β-values were converted to M-values (log_2_ ratio [methylated signal intensity/unmethylated signal intensity]) to account for heteroscedasticity and allow for analyses assuming a Gaussian distribution [27]. M-values have a range of −∞ to +∞, with a value close to 0 indicating similar intensities between methylated and unmethylated probes. Positive M-values represent hyper-, while negative M-values represent hypo-methylation. M-values were then standardized (converted to Z-scores) to perform hierarchical clustering, using Euclidean distance and average linkage criteria for visualization of methylation signatures. 

### 4.5. Functional Enrichment Analysis

All differentially methylated CpG sites were annotated to genes using the reference sequence database (RefSeq) build 87 and were subjected to functional analysis using KEGG pathway analysis and GO grouping categories (biological process, cellular component, and molecular function). The results of enriched pathways were ranked by enrichment scores to identify overrepresented pathways and then sorted by factor score to consider those pathways with the most significant *p*-value. A high enrichment score indicates that a significant number of the differentially methylated genes within a pathway are present, while factor score enables comparison of pathways with similar enrichment scores between GDM and non-GDM groups. 

### 4.6. Statistical Analysis

Participant characteristics were tested for normality using the Shapiro-Wilk test in STATA 14 (StataCorp, College Station, USA). Normally distributed data are expressed as the mean ± standard error of the mean (SEM), or as the median and interquartile range (25th and 75th percentiles) for data that were not normally distributed. An unpaired *t*-test or the Mann–Whitney test was used to compare variables across GDM groups. Categorical variables were analysed using the chi-square test or the Fisher’s exact test if the frequency was <5. A *p* ≤ 0.05 was considered statistically significant. Due to the matched case control study design, a two-way analysis of variance (ANOVA, one factor was the GDM status and the other was the pairing ID), was used to identify differentially methylated sites. To investigate the association between GDM and differentially methylated CpGs, univariate and multivariate generalised linear regression models were tested and adjust for confounding factors. Pearson’s rank correlation (r) was used to evaluate the relationship between specific CpG DNA methylation (β-values; 0–1, as a percentage of methylated to unmethylated) states and clinical parameters. Pathway enrichment was based on the current publicly available human database, GRCh38, and statistical significance was calculated using Fisher’s exact test. An enrichment score ≥3 was considered significant (*p* < 0.05).

## Figures and Tables

**Figure 1 ijms-20-05828-f001:**
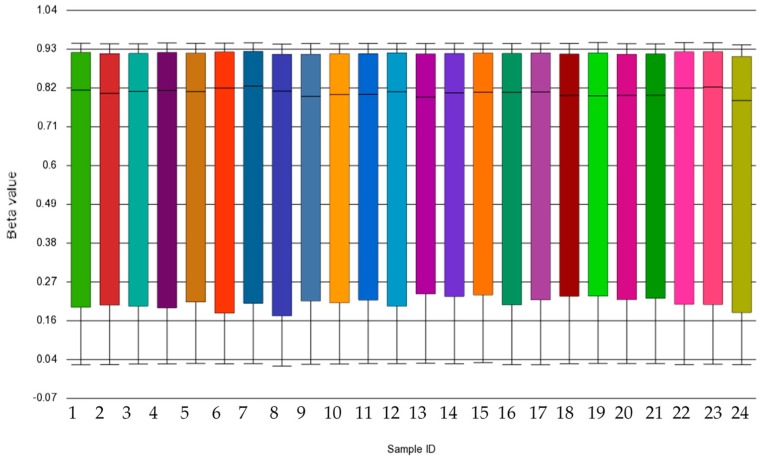
Box and whisker plots of β-values. Each box represents a sample (*n* = 24) which is illustrated by a different color bar. The median β-value is 0.042 with a minimum and maximum range of 0.785 and 0.827.

**Figure 2 ijms-20-05828-f002:**
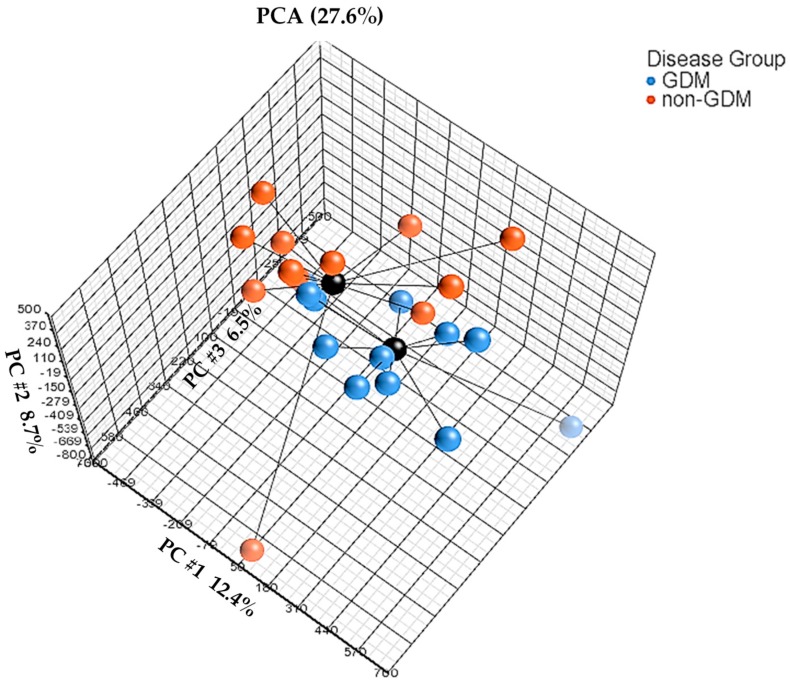
Principal component analysis (PCA) between GDM and non-GDM groups. Each dot represents a sample. Centroids (black) connect samples from the respective GDM (blue) or non-GDM (red) group and indicate the center of distribution, while the black bars indicate the distance between samples and centroids. The first three PCAs explain 27.6% of the variance.

**Figure 3 ijms-20-05828-f003:**
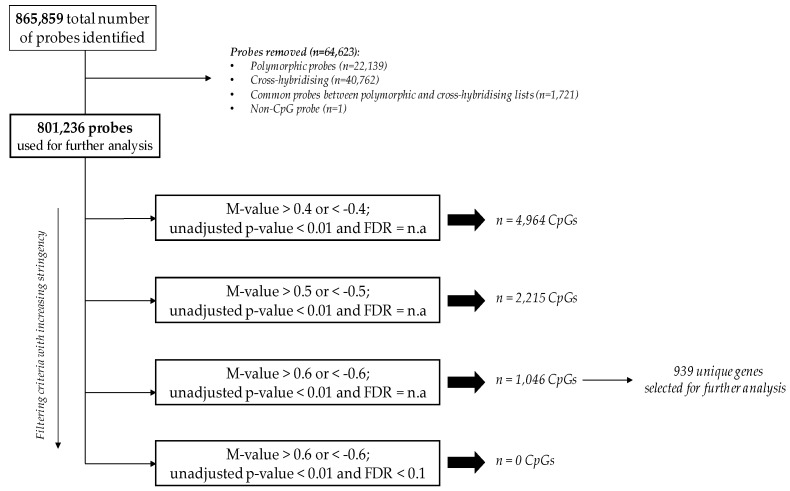
Filtering criteria for the identification of CpGs differentially methylated between GDM and non-GDM groups. A total of 801,236 probes, derived through the removal of polymorphic, cross-hybridising and non-CpG probes were used for analysis. FDR: false discovery rate; M-values closest to 0 indicate similar methylation intensities between probes.

**Figure 4 ijms-20-05828-f004:**
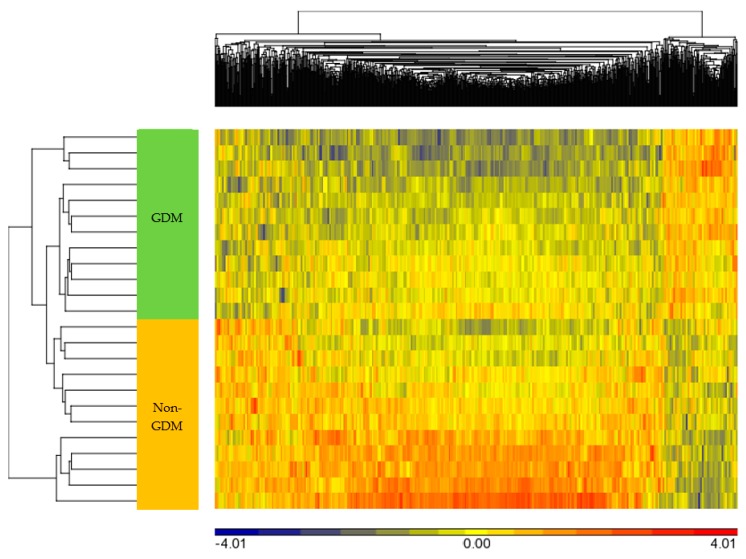
Heatmap showing methylation signatures of 1046 CpG sites in women with/without GDM. DNA methylation across 1046 CpG sites in each sample was analyzed using Euclidean distance for both rows (observations) and columns (features) and average linkage criteria. Samples are shown in rows and are clustered in GDM (green) and non-GDM (orange) groups. Standardized M-values are depicted using a blue (hypomethylation in GDM) to red (hypermethylation in GDM) methylation gradient.

**Figure 5 ijms-20-05828-f005:**
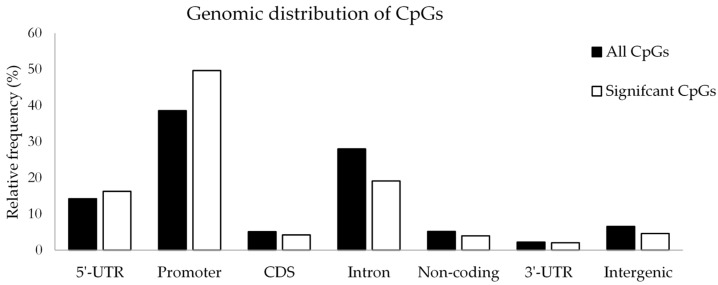
Relative frequency of all CpGs analysed (black bars) and differentially methylated CpGs identified in our study (white bars) in relation to genomic location across the genome. UTR: untranslated region; CDS: coding domain sequence.

**Figure 6 ijms-20-05828-f006:**
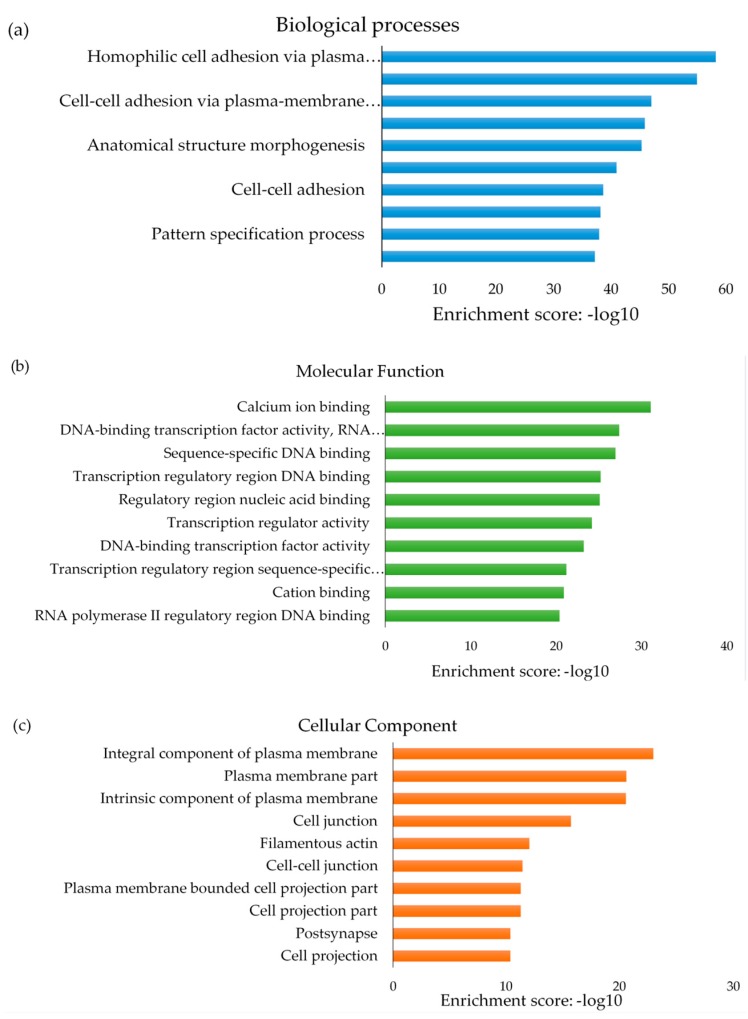
Top 10 Gene Ontology (GO) terms enriched by differentially methylated genes in GDM and non-GDM groups. Enriched GO terms were categorized into (**a**) biological processes, (**b**) molecular function and (**c**) cellular components. Data are presented as enriched scores expressed as −log10 (*p* value). Fisher *p* ≤ 0.001.

**Table 1 ijms-20-05828-t001:** Participant characteristics.

Variables		Non-GDM (*n* = 12)	GDM (*n* = 12)	*p*-Value
Age (years) ^a^		27.3 (0.3)	27.3 (0.3)	1.00
Gestational age (weeks) ^a^		19.3 (1.5)	19.3 (2.0)	1.00
BMI (kg/m^2^) ^a^		27.1 (1.3)	27.6 (1.1)	0.77
Fasting glucose (mmol/L) ^a^		4.3 (0.1)	5.5 (0.1)	<0.001
1hr OGTT (mmol/L) ^a^		5.2 (0.3)	6.6 (0.4)	0.01
2hr OGTT (mmol/L) ^a^		5.2 (0.3)	5.8 (0.3)	0.07
HbA1c (%) ^a^		5.1 (0.1)	5.1 (0.1)	0.85
Fasting insulin (mIU/L) ^b^		8 (7.5-9.0)	10.2 (6.3-12.7)	0.65
HOMA ^b^		1.6 (1.6-1.8)	2.6 (1.5-2.9)	0.31
Adiponectin (µg/mL) ^b^		10.4 (7.3-23.8)	9.7 (4.7-12.0)	0.28
C-reactive protein (mg/L) ^a^		7.1 (1.2)	7.7 (1.1)	0.75
Risk factors: *n* (%) ^c^	None	10 (83.3)	7 (58.3)	0.37
	≥1 risk factor	2 (16.7)	5 (41.8)	
* Education: *n* (%) ^c^	<grade 12	7 (63.6)	5 (41.7)	0.29
	≥grade 12	4 (36.4)	7 (58.3)	
Employment: *n* (%) ^c^	None	8 (66.7)	7 (58.3)	1.00
	Formal/informal employment	4 (33.3)	5 (41.7)	

GDM: gestational diabetes mellitus; BMI: body mass index; OGTT: oral glucose tolerance test; HbA1c: glycated hemoglobin; HOMA: homeostatic model assessment calculated according to the formula: fasting insulin (mIUL) × fasting glucose (mmol/L)/22.5; Risk factors: advanced maternal age (age > 35 years), obesity (BMI > 30 kg/m^2^), family history of diabetes mellitus, delivery of a previous baby more than four kilograms, glucosuria, previous recurrent pregnancy loss, stillbirth, or birth of a baby with congenital abnormalities. * One participant had missing data for education. Data are expressed as the ^a^ mean ± standard error of the mean, as ^b^ median (25th–75th percentiles) or as ^c^ count (percentage). *p*-values for continuous data were calculated using the Mann–Whitney or the unpaired Student *t* test. *p*-values for categorical data were calculated using chi-square test or Fisher’s exact test if frequency was <5.

**Table 2 ijms-20-05828-t002:** The top five significantly differentially methylated CpG sites between GDM and non-GDM groups.

Probe ID	Location	Gene Symbol	Gene Name	Region	*p*-Value	Methylation
cg22985016	Chr5:492187–524227	*SLC9A3*	Solute Carrier Family 9 Member A3	Intron	1.84 × 10^−7^	↑
cg21910650	Chr6:42976841–42986722	*MEA1*; *KLHDC3*	Male-Enhanced Antigen 1; Kelch domain-containing protein 3	Promoter/5’UTR	3.23 × 10^−6^	↓
g23643951	Chr1:7151432–7309551	*CAMTA1*	Calmodulin Binding Transcription Activator 1	Intron	4.46 × 10^−6^	↓
cg16306629	Chr8:119121060–119129059	*COLECT10* *	Collectin Subfamily member 10*	Enhancer *	9.22 × 10^−6^	↑
07966372	Chr13:114782770–114898099	*RASA3*	RAS P21 Protein Activator 3	5’UTR/Intron	9.75 × 10^−6^	↓

* Nearest gene/regulatory region of cg16306629. ↑: hypermethylation and ↓: hypomethylation between GDM vs. non-GDM groups. Significance is shown as *p* < 0.05.

**Table 3 ijms-20-05828-t003:** Linear regression analysis of gestational diabetes mellitus and the top five significantly differentially methylated CpG sites, adjusting for age, body mass index and gestational age.

CpG Site	^a^ Univariate			^b^ Multivariate		
	Coefficient	95% CI	*p*-Value	Coefficient	95% CI	*p*-Value
cg22985016 *(SLC93A)*	0.028	0.019; 0.037	<0.001	0.028	0.019; 0.037	<0.001
cg21910650 *(MEA1;KLHDC3)*	−0.088	−0.117; −0.058	<0.001	−0.087	−0.118; −0.056	<0.001
cg23643951 *(CAMTA1)*	−0.056	−0.070; −0.042	<0.001	−0.056	−0.071; −0.042	<0.001
cg16306629 *(Unknown)*	0.274	0.183; 0.366	<0.001	0.275	0.192; 0.359	<0.001
cg07966372 *(RASA3)*	−0.015	−0.025; −0.004	0.006	−0.015	−0.026; −0.004	0.008

^a^ Univariate linear regression: association between CpG-specific methylation and GDM. ^b^ Multivariate linear regression: adjusting for age (years), body mass index (kg/m^2^) and gestational age (weeks); CI: Confidence interval. Significance is shown as *p* < 0.05.

**Table 4 ijms-20-05828-t004:** Correlation analysis showing the association between DNA methylation and fasting plasma, 1 h OGTT, 2 h OGTT and fasting insulin for the top five differentially methylated CpG sites.

Variable	cg22985016 (SLC93A)		cg21910650 (MEA1; KLHDC3)		cg23643951 (CAMTA1)		cg16306629 (Unknown)		cg07966372 (RASA3)	
	Rho	*p*-Value	Rho	*p*-Value	Rho	*p*-Value	Rho	*p*-Value	Rho	*p*-Value
Fasting glucose (mmol/L)	0.728	<0.001	−0.694	<0.001	−0.735	<0.001	0.724	<0.001	−0.452	0.026
1 h OGTT (mmol/L)	0.502	0.012	−0.377	0.069	−0.399	0.053	0.559	0.004	0.016	0.939
2 h OGTT (mmol/L)	0.297	0.168	−0.249	0.250	−0.338	0.115	0.266	0.219	0.098	0.658
Fasting insulin (mIU/L)	−0.037	0.888	−0.103	0.691	−0.204	0.433	0.109	0.674	−0.495	0.043

OGTT: oral glucose tolerance test; SLC93A: Solute Carrier Family 9 Member A3; MEA1; KLHDC3: Male-Enhanced Antigen 1; Kelch domain-containing protein 3; CAMTA1: Calmodulin Binding Transcription Activator 1; Unknown: gene nearest to this region is called Collectin Subfamily member 10; RASA3: RAS P21 Protein Activator 3. Pearson’s correlation coefficient (rho) is shown with significance at *p* < 0.05.

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
