# Peer review of "Altered Genome-Wide DNA Methylation in Peripheral Blood of South African Women with Gestational Diabetes Mellitus"

_ijms, 2019, doi:10.3390/ijms20235828_

Round 1
Reviewer 1 Report
Dias et al. investigate the methylation profile in women with or without gestational diabetes mellitus (GDM) in an EPIC array based EWAS. Their motivation is to identify potentially screening markers for GDM as GDM is associated with varies morbidities. An interesting finding was made in the CAMTA1 gene known for its involvement in insulin regulation. The manuscript is well written and structured and of general interest for researchers in the field of GDM and diabetes in general.
I have the following comments:
In conjunction with figure 1 a density plot should be shown (e.g. as supplementary) as distribution is hidden in box plots. Page 2, line88; “risk factors” should be explained in greater detail. Figure 1; the range of medians appears much narrower than the stated 0.7-0.9 Figure 5; for comparison, the authors should include bars corresponding to all CpGs on the EPIC array stratified by the six categories. How did the authors perform correction for multiple testing? Was this prior or post size effect filtering? If the latter this in incorrect, the FDR has to be based on all the app 813,135 tests. Page 6, line 148; the associated sites of the two probes should be elaborated, i.e. where in CAMTA1/unknown gene is the probe located and what is the nearest gene/regulatory region of the unknown gene. Page 6, line 156-157; how can the association between GDM and cg23643951 be too strong for logistic regression? Figure 6, A-F indications are missing. The authors could with advantage perform meta-analysis with e.g. the results from Kang et al. or at least illustrate the degree of overlap in a figure/table with direction of effect p-value etc. Page 9, line 226-227, the larger than sign should be flipped. Page 10, line 281-282, the statement concerning IGF-2 is not supported elsewhere in the manuscript. Header in 4.2. and 4.3. is identical Page 12, line 362, why did the authors remove sex chromosomes when all study subjects are female? The authors should provide more information on probe removal especially concerning cross-hybridization and SNP probes - a total removal of only app. 33k probes sounds very low. Did the authors look at the covariates specifically for DMPs found? It is conceivable that the methylation status of e.g. cg24513014 is significantly different in the blood cell types and therefore cell type composition should be corrected for. The authors should perform a multivariate logistic regression with all filtered probes - including cell type composition. Include QQ plots Authors should provide more details on their power calculations - what exact values were used and does the power match their findings?Author Response
Response to reviewer 1 comments:
Point 1: In conjunction with figure 1 a density plot should be shown (e.g. as supplementary) as distribution is hidden in box plots.
Response 1: Thank you for the comment. A density plot illustrating distribution of methylation beta values in all samples has been added as a supplementary Figure S2, as suggested. The results have briefly been described on page 3, line 111-112 and have been mentioned in the methodology on page 14, line 446-447.
Point 2: Page 2, line88; “risk factors” should be explained in greater detail.
Response 2: Risk factors for GDM have been defined in the footnote of Table 1 (page 3). However, in response to the reviewer’s comments and to clarify, risk factors have also been described in greater detail in the results section on page 2, line 89-92, and have been added to the methodology on page 13, line 391-393.
Point 3: Figure 1; the range of medians appears much narrower than the stated 0.7-0.9
Response 3: To clarify, we have given the min and max median beta values across the 24 samples. This has been added to the results, on page 3, line 109-110.
Point 4: Figure 5; for comparison, the authors should include bars corresponding to all CpGs on the EPIC array stratified by the six categories.
Response 4: Figure 5 has been edited to include the genomic distribution of all CpGs on the EPIC array that have been analyzed: page 7.
Point 5: How did the authors perform correction for multiple testing? Was this prior or post size effect filtering? If the latter, this in incorrect: the FDR has to be based on all the app 813,135 tests.
Response 5: The FDR was based on the appropriate number of tests as the reviewer has proposed. The filtering criteria in Figure 3 has been amended to clarify this in the manuscript on page 5. Importantly, also note that the data has been re-analyzed in response to the second reviewer’s comments which has changed the number of probes identified (from 813,135 to 801,236). All further analysis was based on the 801,236 probes.
Point 6: Page 6, line 148; the associated sites of the two probes should be elaborated, i.e. where in CAMTA1/unknown gene is the probe located and what is the nearest gene/regulatory region of the unknown gene.
Response 6: The data was re-analysed, and in addition to CAMTA1 there are other probes that are significant, which has been included in Table 2 indicating location of the probe, gene symbol, gene name, region, p-value and direction of methylation between GDM vs non-GDM groups. In addition, the nearest gene/regulatory region has been added for the probe associated with the unknown gene: Page 8
Point 7: Page 6, line 156-157; how can the association between GDM and cg23643951 be too strong for logistic regression?
Response 7: Thank you for the comment. We apologize for the incorrect phrasing in this section. Subsequently, we have consulted a biostatistician who recommended we perform a generalized linear regression analysis to determine the association between methylation at the specific CpG sites identified and GDM. In light of these comments, we performed a univariate and multivariate linear regression adjusting for age, BMI and gestational age. The results have been described on page 7, line 171-172 and are presented in Table 3 (page 8).
Point 8: Figure 6, A-F indications are missing.
Response 8: Thank you for bringing this to our attention. Following the re-analysis of our data, additional significant CpG sites have been reported on. These results have been presented in the form of tables (Table 3 and 4) on page 8 and 9, instead of figures.
Point 9: The authors could with advantage perform meta-analysis with e.g. the results from Kang et al. or at least illustrate the degree of overlap in a figure/table with direction of effect p-value etc.
Response 9: Thank you for this insightful comment. We have subsequently contacted the authors to ask if they would be willing to release their data for a meta-analysis, but we are still awaiting a response. We were, however, able to show the degree of overlap for differentially methylated genes identified between our study and that of Kang et al. from their available supplementary data. This has been illustrated in a venn diagram and has been added as a supplementary Figure S3, and discussed on page 11, line 279-287. In addition, the direction of methylation (hyper- or hypo-methylation) has been indicated for genes in GDM vs. non-GDM in our study. Unfortunately, this information was not available for Kang et al.
Point 10: Page 9, line 226-227, the larger than sign should be flipped.
Response 10: Thank you for bringing this to our attention, the larger than sign has been changed to < in the discussion: page 11, line 291-292.
Point 11: Page 10, line 281-282, the statement concerning IGF-2 is not supported elsewhere in the manuscript.
Response 11: Thank you for the comment. In an attempt to keep the manuscript focused, the section on IGF has been removed from the discussion: Page 12, line 343-345 and 347-350.
Point 12: Header in 4.2. and 4.3. is identical Page 12,
Response 12: We apologize for the typing error. Header 4.3 has been corrected and changed to: ‘Genome-wide DNA methylation profiling’ on page 14, line 420.
Point 13: Line 362, why did the authors remove sex chromosomes when all study subjects are female?
Response 13: Thank you for this comment, the data has been re-analyzed with all the chromosomes, and this sentence has been omitted from the methodology. Page 14, line 440-441.
Point 14: The authors should provide more information on probe removal especially concerning cross-hybridization and SNP probes - a total removal of only app. 33k probes sounds very low. Did the authors look at the covariates specifically for DMPs found?
Response 14: In light of this comment, the data has been re-analysed, and polymorphic, cross-hybridising and non-specific probes have been removed based on the list of probes published by McCartney et al. 2016. In total, 64,623 probes have been removed which include: polymorphic probes with a MAF>=0.05 in at least one of the populations tested (n=22,139), cross-hybridising probes (n=40,762), non-CpG probes (n=1) and probes that were common in the list of polymorphic and cross-hybridising probes (n=1,721). The manifest file contained 865,859 probes, - 64,623 probes removed = a final total of 801,236 used for statistical analysis. This has been described in the methodology on page 14, line 447-450.
Point 15: It is conceivable that the methylation status of e.g. cg24513014 is significantly different in the blood cell types and therefore cell type composition should be corrected for. The authors should perform a multivariate logistic regression with all filtered probes - including cell type composition. Include QQ plots.
Response 15: We found no significant difference between cell type proportion between GDM and non-GDM groups, thus did not adjust for cell type, particularly given our small sample size. However, this is a limitation of the study and is discussed on page 13, line 367-371.
Point 16: Authors should provide more details on their power calculations - what exact values were used and does the power match their findings?
Response 16: The power analysis was calculated using the means and standard deviation of 2hr OGTT, a marker commonly used to characterise GDM and non-GDM groups, from a previous epigenetic study showing significant differential methylation between groups. The analysis showed that a sample size of n=6 was sufficient to detect a statistically significant difference at 80% power. Our sample size of 24 (n=12 per group) estimated a power of 99% and was therefore sufficient to detect a statically significant difference between GDM and non-GDM groups. However, this analysis may not be appropriate for DNA methylation analysis, as observed in our study, and is discussed on page 12, line 362-366 as a limitation. We therefore decided to omit this from the manuscript, page 15, line 491-493.
Reviewer 2 Report
This study investigated the DNA methylation associated with GDM, and reported some interesting results. The paper is well and clearly written, could be an example for how a scientific paper should be written. My major concerns are on the methods.
Do all 24 samples passed the sample quality control? I understand that this is a matched case control design; however, none of the statistical methods have taken this design into consideration. Methylation M-value is a continuous variable. How can a logistic regression be applied to a continuous variable? I do not think this analysis is correct. I do not see much meanings of using quantile regression, especially given the sample size is so small. If the authors would like to estimate the association for cg24513014 and cg23643951, a linear regression could be used. In Figure 3, is the group difference based on raw M-value or standardized M-value? Page 3, the statement "The overall PCA precentage was 27.6%" is wrong. From the methodology of PCA, the overall PCA explain 100% variation. I think it should be the first 3 PCAs explain 27.6% here. Figure 6 suggests that the association between methylation and fasting glucose, 1hr OGTT could be due to confounding of GDM. GDM should be adjusted for in this analysis. The authors stated that they have considered the cell types. However, it is not true. Page 13 the method said that the analysis was not adjusted for cell composition at all. No multiple test adjustment was done. Given the sample size, I do not think there would be meaningful results if the adjustment was done. The paper should discuss this issue.Author Response
Response to reviewer 2 comments:
Point 1: Do all 24 samples passed the sample quality control?
Response 1: In addition to the box and whisker plots which shows concordance across samples, the average detection p-values for all probes were below the usual cut-off of 0.01 indicating that all samples passed quality control. This data is presented as supplementary Figure S1, and is indicated on page 3, line 105-107.
Point 2: I understand that this is a matched case control design; however, none of the statistical methods have taken this design into consideration.
Response 2: Thank you for the comment. The data has been re-analysed using a two-way analysis of variance ANOVA (one factor was the GDM status and the other was the pairing ID) to identify differentially methylated sites. Subsequently, all statistical analysis was based on this data. This has been corrected in the methodology, page 15, line 484-486.
Point 3: Methylation M-value is a continuous variable. How can a logistic regression be applied to a continuous variable? I do not think this analysis is correct. I do not see much meanings of using quantile regression, especially given the sample size is so small. If the authors would like to estimate the association for cg24513014 and cg23643951, a linear regression could be used.
Response 3: A logistic regression was performed to test the association between GDM (dependent binary variable) and M-values (continuous variable). Subsequently we have consulted a biostatistician who recommended we perform a generalized linear regression analysis using methylation as the dependent variable, as suggested. We performed a univariate and multivariate linear regression adjusting for age, BMI and gestational age. This has been corrected in the methodology on page 15, line 486-488, and the results have been described on page 7, line 171-172 and are presented in Table 3 on page 8.
Point 4: In Figure 3, is the group difference based on raw M-value or standardized M-value?
Response 4: Statistical analysis was performed on raw M-values, while standardized M-values were used for the heat map (Figure 4). In addition, Figure 3 has been amended for clarity. Page 5 and 6.
Point 5: Page 3, the statement "The overall PCA percentage was 27.6%" is wrong. From the methodology of PCA, the overall PCA explain 100% variation. I think it should be the first 3 PCAs explain 27.6% here.
Response 5: Thank you for bringing this to our attention. The sentence describing PCA plots has been rephrased in the results section on page 3, line 113-115 and the figure legend of the PCA plot (Figure 2) has been corrected: page 5
Point 6: Figure 6 suggests that the association between methylation and fasting glucose, 1hr OGTT could be due to confounding of GDM. GDM should be adjusted for in this analysis.
Response 6: Thank you for the comment. A univariate and multivariate linear regression analysis has been performed to adjust for GDM. This is discussed in the results section on page 7, line 186-189, and has been added as a supplementary Table S3.
Point 7: The authors stated that they have considered the cell types. However, it is not true. Page 13 the method said that the analysis was not adjusted for cell composition at all. No multiple test adjustment was done. Given the sample size, I do not think there would be meaningful results if the adjustment was done. The paper should discuss this issue.
Response 7: We found no significant difference between cell type proportion between GDM and non-GDM groups, thus did not adjust for cell type, particularly given our small sample size. However, this is a limitation of the study and is discussed on page 12-13, line 367-371.
Round 2
Reviewer 1 Report
The authors have satisfactorily addressed the listed questions and comments
Reviewer 2 Report
The authors have appropriately addressed my concerns.